



# Interannual sedimentary effluxes of alkalinity in the southern North Sea: Model results compared with summer observations

Johannes Pätsch[a,*], Wilfried Kühn[a], Katharina D. Six[b]

[a]*Institute of Oceanography, University of Hamburg, Germany*
[b]*Max-Planck Institute for Meteorology, Hamburg, Germany*

## Abstract

For the sediments of the central and southern North Sea different sources of alkalinity generation are quantified by a regional modelling system for the period 2000 - 2014. For this purpose a formerly global ocean sediment model coupled with a pelagic ecosystem model is adopted to shelf sea dynamics where much larger turnover rates than in the open and deep ocean occurs. To track alkalinity changes due to different nitrogen-related processes the open ocean sediment model was extended by the state variables particulate organic nitrogen (PON) and ammonium. Directly measured and from Ra isotope flux observation derived alkalinity fluxes from the sediment into the pelagic are reproduced by the model system but calcite building and calcite dissolution are underestimated. Both fluxes cancel out in terms of alkalinity generation and consumption. Other simulated processes altering alkalinity in the sediment like net sulfate reduction, denitrification, nitrification and aerobic degradation are quantified and compare well with corresponding fluxes derived from observations. Most of these fluxes exhibit a strong positive gradient from the open North Sea to the coast where large rivers drain nutrients and organic matter. Atmospheric nitrogen deposition shows also a positive gradient from the open sea towards land and supports alkalinity generation in the sediments. An additional source of spatial variability is introduced by the use of a 3D-heterogenous porosity field. Due to realistic porosity variations (0.3 - 0.5) the alkalinity fluxes vary by about 4 %. The strongest impact on interannual variations of alkalinity fluxes exhibit the temporal varying nitrogen inputs from large rivers directly governing the nitrate concentrations in the coastal bottom water, thus, provide nitrate necessary for benthic denitrification. Over the time investigated the alkalinity effluxes decrease due to the decrease of the nitrogen supply by the rivers.

## 1. Introduction
Alkalinity generation from anaerobic degradation in coastal sediments favors the marine uptake
capacity for atmospheric $CO_2$. This is because these paths of organic matter degradation include

---

*Corresponding author
Email address:* johannes.paetsch@uni-hamburg.de (Johannes Pätsch)





irreversible processes like $N_2$ production and loss of reduced sulfate products.
In September 2011 and June 2012 Brenner et al. (2016) measured alkalinity fluxes from the North
Sea sediment using several sediment cores. For the southern North Sea they found a mean flux
of 6.3 $mmol\,m^{-2}\,d^{-1}$. Alkalinity effluxes into the pelagic system could partly determine the rel-
ative high surface alkalinity concentrations (Fig. 1a) in the southern North Sea as observed by
Thomas et al. (2009) in September 2001. Together with observed concentrations of dissolved inor-
ganic carbon (DIC) (Bozec et al., 2006) these surface alkalinity concentrations can be translated
into $\Delta pCO_2$ values ($pCO_2^{ocean} - pCO_2^{atmosphere}$) which are mainly responsible for the air sea ex-
change of $CO_2$ between ocean and atmosphere (Fig. 1b). In the southern North Sea positive
values indicate oversaturation and thus outgassing, whereas in the northern parts negative values
result in an uptake of atmospheric $CO_2$. When in a simple gedankenexperiment the observed
alkalinity fluxes by Brenner et al. (2016) would be reduced by 50 % from the beginning of the
year, the alkalinity concentrations especially in the shallow southern North Sea would be reduced
(Fig. 1c) and the corresponding $\Delta pCO_2$ values would exhibit much stronger oversaturation (Fig.
1d). This simple experiment ignores the seasonality of the alkalinity fluxes and the fact that DIC
fluxes would vary in concert.
In this paper we investigate the variability of alkalinity generation and the efflux to the pelagic
zone by means of a regional biogeochemical model. The second chapter presents methods concern-
ing the model setup, particular with regard to the adaptation of the former open ocean sediment
model (Heinze et al., 1999) to shelf sea conditions. Within the third chapter model results are
compared with observational data. In the fourth chapter we show the results of several scenarios
demonstrating the sensitivity of the total model dynamics on environmental settings due to chang-
ing alkalinity fluxes. One of these scenarios picks up the gedankenexperiment mentioned above.
It demonstrates the strong impact of reduced alkalinity fluxes on the $pCO_2$ (see Chapter 4.2).

## 2. Methods

The simulations were performed with the ecosystem model ECOHAM (Pätsch and Kühn, 2008)
using the nesting method focussing on the central and southern North Sea ($50.88^o$ to $57.28^oN$,
$3.42^oW$ to $9.25^oE$) (Pätsch et al., 2010). The model system includes the hydrodynamic model
HAMSOM (Backhaus, 1985; Pohlmann, 1996; Pätsch et al., 2017) and the vertically resolved sed-
iment model originally developed for the deep open ocean (Heinze et al., 1999). The latter model
has been adopted to shelf sea dynamics, details are discussed below. The 3D fields of temperature
(T), salinity (S), advective flow and vertical turbulent mixing coefficients calculated by HAMSOM
are used as forcing for ECOHAM. The time step of ECOHAM is 5 minutes.





### 2.1. The hydrodynamic Model

The 3D fields of temperature, salinity, advective flow and vertical turbulent mixing coefficients calculated by the hydrodynamic model HAMSOM are used as forcing for the pelagic biogeochemical model ECOHAM. HAMSOM is a baroclinic, primitive equation model using the hydrostatic and Boussinesq approximations. The current velocities are calculated using a first order component-upstream scheme. The horizontal is discretized on a staggered Arakawa C-grid (Arakawa and Lamb, 1977) with a resolution of $\Delta\lambda = 1/3^o$ and $\Delta\phi = 1/5^o$.

In a first step the model was applied on a larger area including the Northwest European Shelf ($15.250^oW$ - $14.083^oE$, $47.583^oN$ - $63.983^oN$) (Lorkowski et al., 2012). For this large-domain run sea surface elevations of the semi-diurnal lunar tide M2 were prescribed at the open boundaries (Backhaus, 1985). The corresponding results of temperature, salinity and surface elevation were stored on the boundaries of the smaller model domain used in this study (black lines in Fig. 2). These data were used as boundary conditions for the hydrodynamic model HAMSOM implemented on the smaller domain with a vertical resolution of 5 m in the upper 50 m and increasing resolution below. The M2-tide is thus induced implicitly by the prescribed surface elevation at the boundaries. Details of the nesting procedure can be found in Schwichtenberg (2013).

### 2.2. The pelagic biogeochemical Module

In the same way as for the hydrodynamic model in a first step the biogeochemical model ran on the larger model domain and provided boundary conditions for the model on the smaller grid (Fig. 2).

The pelagic biogeochemical model includes 4 nutrients (nitrate, ammonium, phosphate, silicate), two phytoplankton groups (diatoms and flagellates), two zooplankton groups (micro- and meso-zooplankton), bacteria, two fractions of detritus (fast and slowly sinking), labile dissolved organic matter, semi-labile organic carbon, oxygen, calcite, dissolved inorganic carbon, and total alkalinity. Calcite formation is performed by flagellates, only. The molar ratio of soft tissue production to calcite production is 10:1. The model differentiates between normal exudation by phytoplankton, the result of which is labile dissolved organic matter with Redfield composition corresponding to the Redfield production, and an excess exudation of semi-labile organic carbon. The pelagic module is described in detail in Lorkowski et al. (2012). For this study we included the prognostic alkalinity calculation from Schwichtenberg (2013). The different processes (Fi) and their influence on alkalinity are:

- F01 - calcite dissolution

- F02 - calcite formation

- F03 - nitrification



• F04 - uptake of nitrate
• F05 - release of ammonium
• F06 - uptake of ammonium
• F07 - atmospheric deposition of ammonium
• F08 - atmospheric deposition of nitrate
• F09 - uptake of phosphate
• F10 - release of phosphate
These fluxes determine the change of alkalinity:

$$\frac{\partial TA}{\partial t} = 2(F01 - F02 - F03) + F04 + F05 - F06 + F07 - F08 + F09 - F10 \qquad (1)$$



Together with the dynamic sediment module which exchanges TA and DIC with the pelagic system
it was possible to simulate the full carbonate system prognostically.
*2.3. The Sediment Module*
*2.3.1. The open ocean sediment model*
The original sediment model was developed by Heinze et al. (1999) for the global ocean. This
model simulated accumulation, degradation and burial of particulate organic and shell material
and a diffusive pore water exchange with the overlaying ocean. It was applied mainly for the
deep ocean with its low amounts of incoming particulate matter compared to the shallow shelf sea
export. The corresponding time scales of flux variations were rather large (annual to decadal) and
showed no seasonal signal. This model included the solid components particulate organic matter
(POM), calcite, opal and silt exported from the pelagic and the dissolved components phosphate
($PO_4$), dissolved inorganic carbon (DIC), alkalinity (TA), silicate ($Si(OH)_4$), nitrate ($NO_3$), oxy-
gen ($O_2$), and dinitrogen ($N_2$).

*2.3.2. The vertical resolution*
The upper 156 mm of the sediment are resolved by 12 layers with increasing thickness (2 - 24
mm). Below the deepest layer a dimensionless burial layer is implemented.



### 2.3.3. New Components

As the pelagic model delivers sinking particulate material with freely varying stoichiometry, we differentiated benthic POM into the state variables particulate organic carbon (POC), nitrogen (PON) and phosphorus (POP). Additionally we added ammonium ($NH_4$) as product of the incomplete aerobic degradation which can be oxidised by nitrification when oxygen is available (Paulmier et al., 2009). Nitrite is not explicitly included. The model combines the effect of sulfate reduction and reoxidation of reduced sulfate compounds as net sulfate reduction (i.e., sulfate reduction minus reoxidation). The different reaction equations including the alkalinity generation are listed in the appendix.

### 2.3.4. Varying Porosity

The effectivity of several sediment reactions depends on the porosity, i.e., the portion of pore water in a given sediment volume. While the global coarse grid sediment model was implemented with a horizontally uniform porosity (Heinze et al., 1999), in the presented shelf application varying porosities were taken into account. The main parts of the North Sea sediments consist of sand, but there are also muddy areas and even rocky areas exist. The different sediment classes are defined by the composition of grains with different diameters. W. Puls kindly delivered us a North Sea wide map of such grain compositions (pers. comm.). As the sediment model uses porosity values ($P$), the different grain size distributions have to be mapped to porosity values. We used the median grain size ($D50$) to calculate the porosity (pers. comm. W. Puls):

$$P_{surf} = min(1, max(0.3, 0.2603 \cdot 1.20325^{D50})) \tag{2}$$

$$D50 = -log_2 d \tag{3}$$

were $d$ is the grain diameter in mm.

The resulting porosity values $P_{surf}$ fall in the range [0.3,1]. Only for rocky sediments the porosity is defined as zero (Fig. 2). According to Heinze et al. (1999) porosities $P(z)$ in deeper layers were defined in relation to the top layer:

$$P(z) = P_{surf} \cdot e^{k_0 \cdot z(m)} \tag{4}$$

For $k_0 = 2.12$ and $P_{surf} = 0.3$ the deepest layer at $z_{k=12} = -0.144\,m$ obtains a value of $P(z_{k=12}) = 0.22$.



| Process | Turnover Rates | Open Ocean | Shelf | unit | Eqn. No |
|---|---|---|---|---|---|
| aerobic degradation | r1 | $1.160 \cdot 10^{-13}$ | $2.000 \cdot 10^{-10}$ | $\frac{m^3}{mmol\, O_2 \cdot s}$ | (7) |
| denitrification | r2 | $1.157 \cdot 10^{-7}$ | $1.736 \cdot 10^{-3}$ | $\frac{1}{s}$ | (8) |
| sulfate reduction | r3 | $1.157 \cdot 10^{-9}$ | $3.472 \cdot 10^{-9}$ | $\frac{1}{s}$ | (9) |
| calcite dissolution | r4 | $1.000 \cdot 10^{-13}$ | $1.000 \cdot 10^{-8}$ | $\frac{m^3}{mmol\, CO_3^{2-} \cdot s}$ | (10) |
| opal dissolution | r5 | $1.000 \cdot 10^{-12}$ | $1.000 \cdot 10^{-11}$ | $\frac{m^3}{mmol\, SiO_2 \cdot s}$ | (11) |
| nitrification | r6 | | $1.157 \cdot 10^{-4}$ | $\frac{1}{s}$ | (12) |

Table 1: Comparison of open ocean (Heinze et al., 1999) and shelf (this study) turnover rates.

### 2.3.5. Turnover Rates

The reaction equations and the chosen stoichiometries are described in detail in the Appendix. These equations use turnover rates which were modified in comparison to the original open ocean sediment model (see Table 1)

### 2.3.6. Temperature Dependency

As the shallow water column in the North Sea exhibits strong seasonal temperature variations ($\Delta T > 15^o$C) a temperature dependency of both, the turnover rates (see Appendix) and the vertical diffusion was implemented.

A $Q_{10}$ value of 1.2 for aerobic degradation, denitrification, nitrification, sulfate reduction and the dissolution of calcite and opal was chosen (see Appendix).

The vertical diffusion coefficient for all pore water tracers in the original open ocean model was constant (dv $= 10^{-9}\frac{m^2}{s}$). In the shelf model the coefficients were defined as temperature (T) and porosity (P) dependent (Gypens et al., 2008):

$$dv = \begin{cases} (d_0 + a \cdot T) \cdot P & : \quad P < 0.4 \\ (d_0 + a \cdot T) \cdot P^2 & : \quad P \geq 0.4 \end{cases}$$

The parameters $d_0$ and a are defined in Gypens et al. (2008) (their Table 2) for different groups of pore water tracers: The lowest coefficient is defined for phosphate ($dv_{pho}(T_{10}) = 5.4 \cdot P \cdot 10^{-10}\frac{m^2}{s}$) a medium coefficient ($dv_{tra}(T_{10}) = 1.4 \cdot P \cdot 10^{-9}\frac{m^2}{s}$) is valid for the biogeochemical tracers DIC, nitrate, ammonium, alkalinity and silicate. The highest coefficient was defined for the gases oxygen and dinitrogen ($dv_{nit}(T_{10}) = 1.6 \cdot P \cdot 10^{-9}\frac{m^2}{s}$), all at $T_{10} = 10^o$ C and a porosity P $< 0.4$, which is typical for sandy ground. In order to take into account advective exchange of pore water with the pelagic system the coefficients for the uppermost layer were increased by a factor of 10. This factor was determined by several sensitivity runs to balance the exchange between the sediment and the pelagic. The same factor is used by Neumann et al. (2017) to switch between diffusive and



advective nitrate exchange between sediment and pelagic in the German Bight. The temperature
of the sediment was defined as the temperature of the lowest pelagic layer.
The vertical diffusion coefficient for DIC compares well with the corresponding coefficient given
by Burdige and Komada (2013) (their Table 3) for $T = 5^0$ C and $P = 0.36$.

### 2.4. External Data

The meteorological forcing (Kalnay et al., 1996) and the river loads of carbon, alkalinity, nutrients
and organic compounds have been implemented according to Lorkowski et al. (2012). To treat
these tracers more realistically in this study also daily freshwater discharge of the rivers was used
(Pätsch et al., 2016). In this way the input of tracers from the rivers ($mmol\, d^{-1}$) can be an
effective source or sink depending on the concentrations of the tracers in the river water.
The calculated shortwave incoming radiation has been reduced by 10% as it has been shown
that the sea surface temperature (SST) would otherwise be overestimated (compare Fig. 3 in
Lorkowski et al. (2012)).
The atmospheric nitrogen deposition was derived following Große et al. (2016), using annual data
from the EMEP (Cooperative program for monitoring and evaluation of the long-range transmis-
sions of air pollutants in Europe) model. As our simulation period exceeds the period of data
available from EMEP a long-term trend according to Schöpp et al. (2003) was applied in addition.
Atmospheric deposition is implemented as inputs of nitrate and ammonium.

### 2.5. The Experiments

For each experiment described below the biogeochemical simulation in the central and southern
North Sea area spun up over 20 years repeating the year 2000 until all processes were in equlibrium
and did not change from year to year. After this procedure the years 2000 to 2014 were simulated
consecutively.

Different experiments or scenarios were performed:
• The **reference run** with the new sediment module provides a basis with realistic boundary
conditions and horizontally varying porosities.
• In order to reproduce a situation without anthropogenic influence, we reduced the inorganic
and organic river input of nitrogen and phosphorus to 10 % of the reference run. Addition-
ally the atmospheric deposition of nitrogen was reduced to 28 %. This run more or less
reproduced the **"pristine conditions"** Serna et al. (2010) established.





- To analyse the impact of the new sediment module on the pelagic system we compare the results of the reference run with results of the scenario **"plate run"**. In this scenario a simple sediment module was used, which collects, remineralises and releases the sunken particulate organic material on a two dimensional plate (Pätsch and Kühn, 2008).

- In the reference run horizontally varying porosities were used. To study the influence of this feature we conducted two additional model runs with basin wide uniform porosities: One with the minimum porosity $\mathbf{P_{min} = 0.3}$ found in the model domain of the reference run and one with the maximum porosity $\mathbf{P_{max} = 0.51}$.

## 3. Comparison with Observations

To get confidence into the adapted sediment model we compared simulated and observed fluxes between sediment and pelagic. Additionally, simulated pore water profiles were compared with observed profiles.

### 3.1. Oxygen Fluxes

Brenner et al. (2016) measured the total oxygen consumption of sediment cores which can be compared with simulated oxygen fluxes into the sediment. The corresponding available data and their positions are shown in Fig. 3 (rectangles). The underlying map of simulated oxygen fluxes at the time when observations were taken show reasonable values. Only in the German Bight the model underestimates the measurements. An explanation for this effect is that particulate organics (POM) imported by the rivers are considered as slowly sinking detritus. As consequence the horizontal export of POM out of the German Bight is overestimated and the local flux into the sediment is underestimated.

### 3.2. Alkalinity Fluxes

Fig. 4 shows the comparison of fieldwide averaged alkalinity effluxes and the contributions from aerobic degradation, denitrification, net sulfate reduction, nitrification and calcite dissolution from observations in September 2011 (Brenner et al., 2016) and from our model results for September 2011. For the observational data only the spatial standard deviation of alkalinity efflux is given (see grey error bar in Fig. 4a). The temporal standard deviation of the simulated daily values within September 2011 is for all fluxes very small and not shown ($< 0.003 \, \mathrm{mmol \, m^{-2} \, d^{-1}}$). The spatial standard deviation of the simulated September fluxes are shown as error bars in Fig. 4b. Even though the simulated efflux lies within the high spatial variability of the observed alkalinity efflux, the model rather underestimates all contributions. Only the simulated contribution from aerobic degradation is larger than the corresponding observation. The main deviation can be attributed to the low simulated calcite dissolution within the sediment.



In comparison to other models (Ridgwell et al., 2007; Lorkowski et al., 2012) the ratio of simulated particulate organic carbon to particulate inorganic carbon (POC:PIC) or the simulated organic carbon to calcite production ($10\,\mathrm{mol\,POC\,d^{-1}}$ : $1\,\mathrm{mol\,calcite\,d^{-1}}$) is relatively low meaning high calcite production in relation to organic carbon production. Nonetheless our model still leads to an underestimation of the calcite dissolution in the sediment compared to the analysis of Brenner et al. (2016). The mean simulated calcite deposition into the sediment in September 2011 was $2.9\,\mathrm{mmol\,m^{-2}\,d^{-1}}$, which is still only 54 % of the observed calcite dissolution in the sediment.

Fig. 5 shows the corresponding simulated alkalinity fluxes to the pelagic system for 15 September 2011. The alkalinity efflux is strongest in the German Bight near the mouth of River Elbe. The flux decreases with distance from the continental coast. Elevated values can be seen off the Danish coast. Similar features can be observed for the contributors aerobic degradation, denitrification, net sulfate reduction, and calcite dissolution. The distribution of the negative fluxes due to nitrification shows also elevated values in the German Bight.

When ignoring the sedimentary calcite dissolution in both the simulation and the observed data, the remaining alkalinity generation compares better with the observations (Fig. 6). The colors within the four rectangles identify the strength of the observational data whereas the horizontal distribution shown by the colored map stands for simulations. Only in the inner German Bight the simulated flux appears far too low. An explanation for this effect is the same as for oxygen fluxes: The export of POM out of the German Bight is overestimated and thus local remineralization underestimated.

### 3.3. Profiles

During the cruise He-308 in May 2009 several sediment cores in the German Exclusive Economic Zone (EEZ) were taken and investigated. The nitrate data are published by Neumann et al. (2017), all data are archived in Pangaea (2017). We compare our results of the reference run with observed data of oxygen, nitrate, phosphate and ammonium (Fig. 7). To understand the model sensitivity, also the corresponding profiles of the "pristine conditions" run are shown. The position of the chosen core is between the German coast and the island Helgoland ($54^{o}\,5^{'}$N, $8^{o}$E). This area is strongly affected by high nutrient loads from the continental rivers and high atmospheric nitrogen deposition (Pätsch et al., 2010) resulting in signifcant differences in the simulated porewater concentrations of the reference run and the scenario "pristine conditions" (solid and dashed black lines). The simulated oxygen penetration depth (concentration $< 10\,\mathrm{mmol\,O_2\,m^{-3}}$) is about $0.5\,\mathrm{cm}$ which fits to the observations (Fig. 7a). It is about $0.8\,\mathrm{cm}$ in the scenario "pristine conditions". In the upper $0.4\,\mathrm{cm}$ the model underestimates in both scenarios the observed oxygen concentrations. Fig. 7b shows the profiles of observed $NO_x$ including nitrate and nitrite



and the profiles of simulated nitrate. We think that this is a proper comparison as observed nitrite concentrations are low ($< 0.8 \, \mathrm{mmol \, N \, m^{-3}}$, not shown). Observed $NO_x$ concentrations are detectable only in the upper $2 \, \mathrm{cm}$. Due to uncalibrated measurements deeper values appear discriminable from zero concentration, but they should be interpreted as zero concentration (pers. comm. Andreas Neumann). The simulated concentrations (reference run) reach very low values already at $1 \, \mathrm{cm}$ depth, the "pristine conditions" scenario shows very low concentrations already at $0.5 \, \mathrm{cm}$ depth. Observed phosphate concentrations in Fig. 7c indicate two mixing regimes: In the upper $9 \, \mathrm{cm}$ the sediment core shows concentrations slightly increasing with depth, below a stronger gradient can be seen. The upper part appears well-mixed while in the lower part mixing decreases. This effect might be caused by bioturbation and bioirrigation in the upper $9 \, \mathrm{cm}$. As the latter processes are not included in the model we got a more homogenous picture of the phosphate profiles. The model (reference run) overestimates the observational values in the upper part while it underestimates them in the lower part. A similar pattern can be seen for ammonium (Fig. 7d), where again the observational concentrations indicate an upper and a lower mixing zone. The simulated values increase between the surface and the $5 \, \mathrm{cm}$ horizon, below they are more or less constant. The values of the reference run are too high in the upper $13 \, \mathrm{cm}$. These high simulated ammonium values might be caused by neglecting the process of anammox in the model. This process transforms reactive nitrogen compounds (ammonium and nitrite) into inert molecular nitrogen. Similar high ammonium concentrations can be found in Luff and Moll (2004) within their Fig. 9.

## 4. Results

### 4.1. Temporal Variations

The temporal development of monthly alkalinity effluxes (2000 - 2014) without calcite dissolution of a near coast station ($54^o$ N, $8^o$ E) shows an overall decreasing trend (Fig. 8). To understand this feature the sources of alkalinity generation and the annual loads of nitrate by the River Elbe (Radach and Pätsch, 2007; Pätsch et al., 2016) in the German Bight ($53.9^o$ N, $8.9^o$ E) are shown (see Fig. 1a). Calcite dissolution is very variable and exhibits a decrease over the simulation period. Because of its high variability which would overwrite the nitrogen-related signals calcite dissolution is not shown.

Aerobic degradation with a distinct annual cycle appears quite constant over the years. Sulfate reduction is more or less constant, while nitrification (as negative contribution) shows a positive trend in opposite to the negative trend of denitrification. The dark blue line represents the nitrate discharge of the nearby River Elbe. With strong seasonal peaks it exhibits a negative trend which can explain a similar trend in denitrification.





Strong nitrate discharge events are followed some months later by local maxima in denitrification.
This can be seen in 2003 and 2011. Over several successional winter months in 2007/2008 high
nitrate loads lead to strong denitrification in 2008. In all these years the TA efflux was elevated.
*4.2. Alkalinity Generation and $pCO_2$*
As already demonstrated in the gedankenexperiment in the introduction the alkalinity release from
the sediment has a significant impact on the carbonate system and thus on the $\Delta pCO_2$ regulating
the exchange of $CO_2$ between the atmosphere and the sea.
Using the simulated timeseries 2000 - 2014 (reference run) we analysed the cummulative alkalinity
efflux out of the sediment from the beginning of the year 2011 to mid September 2011 (Fig. 9a).
Near the Danish coast we found a flux of about 1000 $mmol\,m^{-2}$ for this period. For the inner
German Bight even higher values can be found. These maxima result in corresponding areas of
strong undersaturation in respect of $\Delta pCO_2$ on September 15 (Fig. 9b). The interior and the
northwestern part are slightly oversaturated.
Fig. 9c shows the alkalinity flux of the "pristine conditions" run until September 15. The flux
reduction (compare with Fig. 9a) is strongest ($\approx$ 20 %) in areas where the generation of alkalinity
was strongest. Areas of oversaturation of $\Delta pCO_2$ (Fig. 9d) increase and especially in the shallow
areas with high sediment impact the previously undersaturated situations turn into oversaturation
(or light undersaturation). Because of the distance to the rivers the situation is more or less
unchanged in the central part.
*4.3. Sensitivity on different porosities*
To investigate the effect of spatially varying porosities we conducted two additional simulations
which were spun up separately : One with a basin wide uniform porosity with the minimum value
of the reference run except for rocks ($P_{min}$ = 0.3) and one with the maximum value ($P_{max}$ = 0.51).
For the different annual fluxes between the sediment and the pelagic at $54\,^o\,5\,'$N, $8\,^o$ E the relative
deviations (%) of these two runs are analysed for 2011 (Fig. 10).
Switching over from the $P_{min}$ to the $P_{max}$ run the diffusive flux of DIC, alkalinity and phosphate
out of the pore water of the sediment increases by about 4 %. The flux of silicate from the
sediment increases by 16 %. Also the import of oxygen and nitrate increase. This overall increase
is astonishing as the effective diffusivity decreases when the porosity passes over the limit of 0.4
(see section 2.3.6). Of interest are also the deviations of the five contributors to the alkalinity flux,
i.e., the alkalinity flux due to the aerobic degradation (+7 %), the calcite dissolution (+3.4 %), the
denitrification (+1.5 %), the sulfate reduction (-7.8 %) and the (negative) nitrification (-0.5 %).
Sulfate reduction decreases as the amount of POC reaching the deeper sediment layers decreases
due to the enhanced aerobic remineralisation.





Due to positive feedback mechanisms on the nutrients in the water column the sinking fluxes of
particulate organic matter (POC, PON, POP) increase. The largest increase in solids entering
the sediment can be found for opal ($+5.6\,\%$) corresponding with the large silicate efflux from the
sediment into the pelagic. Calcite export slightly decreases as the silicon shell building diatoms
are favored by the increased silicate availability.
*4.4. Comparison of the vertical resolved and the plate sediment module*
In former model versions (Pätsch and Kühn, 2008; Lorkowski et al., 2012; Große et al., 2016) the
sediment was represented by a two-dimensional plate without depth resolution. The sinking ma-
terial was gathered and remineralised on the surface of this plate. The remineralisation rates had
been adjusted so that the particulate material from the last year was more or less dissolved and
released until February/March of the following year.
The temporal development of carbon exchange between sediment and pelagic in 2011 at $54^o\,5^{'}$N,
$8^o$ E is shown in Fig. 11a for the "plate run". The time in the year when half of the exported
particulate material is returned as DIC ("half time") is indicated by the black arrow on the x-axis.
For the "plate run" this is day 230.
Fig. 11b shows the corresponding carbon fluxes of the reference run. While the shape of the curve
representing the particulate export is similar to that of Fig. 11a, the remineralisation flux shows
less temporal variation. Due to the high remineralisation flux in winter the "half time" is reached
earlier (day 207).

## 5. Discussion


*5.1. Nitrogen related processes*
After calcite dissolution benthic denitrification is the second largest positive contribution to alka-
linity generation (Fig. 4). Near-bottom nitrate concentration which is correlated with near bottom
oxygen saturation governs the direction of nitrate exchange across the pelagic - sediment interface
(Neubacher et al., 2011). In case of the invasion of pelagic nitrate into the sediment benthic den-
itrification is stimulated. The other source of benthic nitrate is the benthic nitrification which is
driven by oxygen within the sediment. At $54^o$ N, $8^o$ E, however 86 % of oxygen are consumed by
aerobic degradation and only 14 % by benthic nitrification. For shelf seas Seitzinger and Giblin
(1996) estimated the local benthic denitrification rate (DNR) to be about 2 % of the local primary
production (PP). This estimate, of course, can be influenced by parameters like water depth, ad-
vection, and near bottom oxygen consumption. Indeed the evaluation of our reference run shows
that the relation r=DNR/PP was about 2 % in regions with water depth of about 30 m and an
annual Redfield production (see 2.2)) of about 150 $g\,C\,m^{-2}\,yr^{-1}$, which can be found some tens
of kilometers off the mouths of the big rivers. According to our simulations r is only larger than





2 % near the mouth of River Elbe. For all other regions r ranges between 1.1 % and 1.4 %.
In the case of the ammonium profile (Fig. 7d) the "pristine conditions" simulation matches the
observation better than the reference run. This might have to do with the absence of the process
anammox within the model which would consume ammonium under presence of nitrite.
The comparison of the reference run and the "pristine conditions" run exhibits a deeper penetration
of oxygen into the sediment for the pristine more nutrient depleted scenario (Fig. 7a). This is in
accordance with the findings of Neubacher et al. (2011) who differentiated a realistic and a rich
hypoxic situation, the latter with lower penetration depths.
*5.2. Sources of Alkalinity*
An effective tracer of North Sea total alkalinity is the naturally occurring radium isotope $^{228}$Ra
(Burt et al., 2014). These authors estimated a coastal alkalinity input of 3.4 - 23.6 $mmol\,m^{-2}\,d^{-1}$
into the southern North Sea (A=190.765 km$^2$) in September 2011. This input was assumed to come
mainly from the Wadden Sea. The amount of this input lies in the same range Brenner et al. (2016)
estimated as total input from the sediments into the pelagic southern North Sea (Fig. 4). This
estimate is valid for a late summer situation and includes the large effect of calcite dissolution. For
the southern North Sea calcite dissolution and production is roughly balanced on an annual basis.
The estimate by Burt et al. (2014) and the measurements by Brenner et al. (2016) appear high
in comparison to the value given by Thomas et al. (2009) who estimated an alkalinity input from
the Wadden Sea of 1 $mmol\,m^{-2}\,d^{-1}$ into the south-eastern North Sea over the year. This value
was calculated using an alkalinity budget which does not differentiate input from autochthonous
sediment and from the adjacent Wadden Sea and, additionally, does not take into account calcite
production and dissolution.
Together with our simulation results the following picture can be given: The relative high flux
esimates by Burt et al. (2014) and Brenner et al. (2016) can be explained by the inclusion of calcite
dissolution and the time in the year when measurements were taken. When calcite dissolution is
excluded our annual estimate for the total model region (0.4 $mmol\,m^{-2}\,d^{-1}$) can be compared with
the estimate by Thomas et al. (2009) for the south-eastern North Sea with higher productivity
than the average of the model region.
**6. Conclusion**
Even though our model may slightly underestimate benthic denitrification in the southern North
Sea it reveals this process as the largest net contribution to alkalinity generation in this area.
This compares well with the estimates by Brenner et al. (2016) when the dissolution of calcite is



not taken into account, because the observational data might miss the calcite production signal
which then would cancel out the effect on alkalinity. Estimates of other alkalinity fluxes like
alkalinity generation in the Wadden Sea are not taken into account as their estimates appear
not well constrained. A direct modelling approach of such sources of alkalinity appears necessary
(Schwichtenberg, 2013), but is beyond the scope of this study.
**7. Acknowledgement**
This work was supported by the Cluster of Excellence CliSAP (EXC177), University of Hamburg,
funded by the German Science Foundation (DFG). We thank Ernst Maier-Reimer who can be im-
mediately identified as coauthor of the model code, Helmuth Thomas, Hermann Lenhart, Alberto
Borges, Markus Kreus, Fabian Schwichtenberg and Fabian Große for valuable discussions.
**8. Figure Caption**
Fig. 1 a) Surface alkalinity concentrations ($\text{mmol kg}^{-1}$) measured in September 2001, b) corre-
sponding $\Delta pCO_2$ (ppm), the difference of partial pressure of ocean and atmospheric $pCO_2$, c)
reduced alkalinity concentrations due to a reduction of 50 % of the estimated alkalinity flux by
Brenner et al. (2016), d) $\Delta pCO_2$ corresponding to the reduced alkalinity in c).

Fig. 2 Porosity field according to W. Puls (pers. comm.). Blue areas indicate rocky sediments, red
areas indicate muddy sediments with low grain diameters and green areas indicate sandy ground.
The black lines indicate the model boundaries.

Fig. 3 Simulated and observed oxygen fluxes ($\text{mmol O}_2 \text{ m}^{-2} \text{ d}^{-1}$) for 15 September 2011. The ob-
servations by Brenner et al. (2016) are indicated by the colored rectangles.

Fig. 4a): Mean observed alkalinity flux for the southern North Sea in September 2011. Addition-
ally the derived mean alkalinity generation due to aerobic degradation, dentrification, net sulfate
reduction and calcite dissolution are shown. A sink for alkalinity is nitrification. All fluxes in
$\text{mmol m}^{-2} \text{ d}^{-1}$ (Brenner et al., 2016). b): Simulated alkalinity flux for the southern North Sea on
15 September 2011. Additionally the alkalinity generation and reduction due to aerobic degrada-
tion, dentrification, net sulfate reduction, nitrification and calcite dissolution are shown. The grey
error bars indicate the spatial standard deviation.

Fig. 5 a) Simulated net alkalinity generation and corresponding sources and sinks due to b) aerobic
degradation, c) denitrification, d) net sulfate reduction, e) nitrification, f) calcite dissolution on



15 September 2011. All fluxes in $\mathrm{mmol\,m^{-2}\,d^{-1}}$. The scale of a) - d) is identical.

Fig. 6 Simulated and observed alkalinity generation ($\mathrm{mmol\,m^{-2}\,d^{-1}}$) without calcite dissolution
for 15 September 2011. The observations are indicated by the colored rectangles.

Fig. 7 Profiles of porewater concentrations of a) oxygen, b) nitrate, c) phosphate and d) ammonium
at $54^{o}\,5^{'}$N, $8^{o}$ E in May 2009. Nitrate data were published by Neumann et al. (2017). The black
solid line indicates the reference run, the dashed black line represents the results of the "pristine
conditions" scenario and the different blue lines are derived from observations during the cruise
He-308. In the figures b-d repeated observational profiles are shown. Notice the different profile
depths.

Fig. 8 Simulated monthly values of alkalinity efflux from the sediment without calcite dissolu-
tion at $54^{o}$ N, $8^{o}$ E, the corresponding sources and sinks due to aerobic degradation, deni-
trification, net sulfate reduction, nitrification and the annual loads of nitrate from River Elbe
(Radach and Pätsch, 2007; Pätsch et al., 2016). Note: nitrification has a negative contribution to
the alkalinity generation.

Fig. 9 a) Simulated cummulative alkalinity generation in 2011 until September 15 ($\mathrm{mmol\,m^{-2}}$)
for the reference run b) the corresponding $\Delta pCO_2$ on September 15, c) cummulative alkalinity
generation until September 15 with reduced river input (10%) and only 28 % atmospheric nitrogen
deposition ("pristine conditions"), d) $\Delta pCO_2$ on September 15 ("pristine conditions").

Fig. 10 Deviations between the "high" and the "low" porosity run. Shown is the relative change
of annual fluxes (%) between the sediment and the pelagic for a station at $54^{o}\,5^{'}$N, $8^{o}$ E in 2011.
DIC, TA, $PO_4$, $SiO_4$, $N_2$, $NH_4$ indicate the diffusive fluxes of dissolved matter from the sediment
into the pelagic. Ox and $NO_3$ are corresponding fluxes from the pelagic into the sediment. aeralk,
cacalk, dnralk, suralk and nitalk indicate the partitioning of the alkalinity flux according to its
sources aerobic degradation, calcite dissolution, denitrification, sulfate reduction and nitrification,
respectively. The fluxes of solids from the pelagic into the sediment are POC, PON, POP, OPAL
and CaCO3.

Fig. 11 Temporal development of carbon fluxes between the pelagic and the sediment at $54^{o}\,5^{'}$N,
$8^{o}$ E in 2011 for a) the "plate run" and b) the reference run. The time in the year when half of the
deposited particulate material is returned as DIC is indicated by the black arrow on the x-axis.



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





## 10. Appendix

*10.1. Equations for solid and dissolved sediment components*

Dissolved Matter: exchange with the pelagic ecosystem model ECOHAM

$(DIC(i = 1), TA(i = 2), PO_4(i = 3), NO_3(i = 4), NH_4(i = 5), N_2(i = 6), SiO_4(i = 7),$
$O_2(i = 8))$

$$\frac{\partial C_i}{\partial t} = dv_i \frac{\partial^2 C_i}{\partial x^2} + \frac{R_j}{P} \qquad j = 1, .., 6 \tag{5}$$

$R_j : [mmol\, m^{-3}\, s^{-1}]$ are reaction terms for the dissolved matter. P: Porosity

$dv_i$ are the vertical diffusion coefficients described in section 2.3.6.

Solids: input from the pelagic ecosystem model ECOHAM

$(POM(C/N/P), CaCO_3, SiO_2$ (Opal), Silt (with predefined constant input))

$$\frac{\partial S_j}{\partial t} = D \frac{\partial(w\, S_j)}{\partial z} - \frac{R_j}{1 - P} \qquad j = 1, .., 5 \tag{6}$$

$$w : \; vertical\, advection\, (downward) \qquad R_j : \; [mmol\, m^{-3}\, s^{-1}]$$

*10.2. Reaction Terms $R_j$*

- *Degradation of POM*

  − aerobic degradation (j=1)

$$R_{POC}^{AD} = r_1 \cdot T_{fac}(T) \cdot [POC] \cdot [O_2] \tag{7}$$

$$r_1 : \; [\tfrac{m^3}{mmol\, O_2 \cdot s}]$$

$$R_{PON}^{AD} = R_{POC}^{AD} \cdot \tfrac{rnit}{rcar}$$

$$R_{POP}^{AD} = R_{POC}^{AD} \cdot \tfrac{1}{rcar}$$

  − anaerobic degradation

  ∗ DNR (denitrification) (j=2)

$$R_{POC}^{DNR} = r_2 \cdot T_{fac}(T) \cdot (1 - \frac{[O_2]}{[O_2^{half}] + [O_2]}) \cdot \min(\frac{\tfrac{1}{2}[NO_3^-]}{nitdem}; [POC]) \tag{8}$$

$$r_2 : \; [\tfrac{1}{s}]$$





* SR (sulfate reduction) (j=3)

$$R_{POCs}^{SR} = r_3 \cdot T_{fac}(T) \cdot \min(\frac{[TA]}{rnit}; [POC]) \qquad (9)$$

$r_3 : [\frac{1}{s}]$
• $CaCO_3\,dissolution\,(j = 4)$

$$R_{CaCO_3} = r_4 \cdot T_{fac}(T) \cdot [CaCO_3] \cdot (\max([CO_3^{2-}]^{sat} - [CO_3^{2-}]; 0)) \qquad (10)$$

$r_4 : [\frac{m^3}{mmol\,CO_3^{2-}\cdot s}]$
$[CO_3^{2-}]^{sat} = \frac{ksp}{Ca}$
ksp: apparent pressure corrected solubility product of calcite
Ca = 10.3 $[mol\,m^{-3}]$: Calcium concentration

• $SiO_2$ (Opal Dissolution) (j=5)

$$R_{SiO_2} = [SiO_2] \cdot r_5 \cdot T_{fac}(T) \cdot ([Si(OH)_4]^{sat} - [Si(OH)_4]) \qquad (11)$$

$r_5 : [\frac{m^3}{mmol\,Si(OH)_4\cdot s}]$
$[Si(OH)_4]^{sat} = 1\,mol\,m^{-3}$
• $NH_4$ (Nitrification) (j=6)

$$R_{NH_4}^{NO_3} = r_6 \cdot T_{fac}(T) \cdot [NH_4] \qquad (12)$$

$r_6 : [\frac{1}{s}]$

$$with\,T_{fac}(T) = 1.2^{\frac{T-T_0}{T_0}}\,with\,T_0 = 10^oC \qquad (13)$$





*10.3. Reaction Equations and Stoichiometry*
All stoichiometric factors are based on $R_0 = rcar + \frac{1}{4}z$      with $rcar = C/P$
and z: H-excess for the notation of organic matter: $C_x\,(H_2O)_w\,(NH_3)_y\,H_z\,H_3PO_4$
1) Incomplete Aerobic Remineralisation (after Paulmier et al., 2009)
$C_a\,H_b\,O_c\,N_d\,P + \underbrace{(a + \frac{1}{4}b - \frac{1}{2}c - \frac{3}{4}d + \frac{5}{4})}_{=ro2ut} \cdot O_2$
$\longrightarrow a \cdot CO_2 + d \cdot NH_3 + H_3PO_4 + (\frac{1}{2}b - \frac{3}{2}d - \frac{3}{2})H_2O$
with $R_0 = a + \frac{1}{4}b - \frac{1}{2}c - \frac{3}{4}d + \frac{5}{4}$    $\Rightarrow$   $ro2ut = R_0$
with $a = rcar = C/P$
$b = \quad\quad = H/P$
$c = \quad\quad = O/P$
$d = rnit = N/P$





2) Complete Denitrification (after Paulmier et al., 2009)
$C_a\,H_b\,O_c\,N_d\,P + \underbrace{(\frac{4}{5}a + \frac{1}{5}b - \frac{2}{5}c + 1)}_{=nitdem} \cdot HNO_3$
$\longrightarrow a \cdot CO_2 + H_3PO_4 + (\frac{2}{5}a + \frac{3}{5}b - \frac{1}{5}c - 1) \cdot H_2O + \underbrace{(\frac{2}{5}a + \frac{1}{10}b - \frac{1}{5}c + \frac{1}{2}d + \frac{1}{2})}_{n2prod} \cdot N_2$
with $R_0 = a + \frac{1}{4}b - \frac{1}{2}c + \frac{5}{4}$    $\Rightarrow$    $n2prod = \frac{2}{5}R_0 + \frac{4}{5}d = \frac{1}{2}(nitdem + d)$
$$nitdem = \frac{4}{5}R_0 + \frac{3}{5}d$$



3) Sulfate Reduction
for $O_2 < 1\mu M$ and $NO_3 < 1\mu M$:
$C_a\,H_b\,O_c\,N_d\,P + \underbrace{(\frac{1}{2}a + \frac{1}{8}b - \frac{1}{4}c - \frac{3}{8}d + \frac{5}{8})}_{=R_0/2} \cdot H_2SO_4$
$\longrightarrow a \cdot CO_2 + d \cdot NH_3 + H_3PO_4 + (\frac{1}{2}b - \frac{3}{2}d - \frac{3}{2}) \cdot H_2O$
$+ \underbrace{(\frac{1}{2}a + \frac{1}{8}b - \frac{1}{4}c - \frac{3}{8}d + \frac{5}{8})}_{=R_0/2} \cdot H_2S$
4) Nitrification of ammonia to nitrate
for $O_2 > 1\mu M$
$NH_3 + 2 \cdot O_2 \longrightarrow HNO_3 + H_2O$
*10.4. Alkalinity Generation*
$R_{TA} = rnit \cdot R_{POC}^{AD} + nitdem \cdot R_{POC}^{DNR} + rnit \cdot R_{POCs}^{SR} + 2 \cdot (R_{CaCO_3} - R_{NH_4}^{NO_3})$



Fig. 1





Fig. 2



Fig. 3

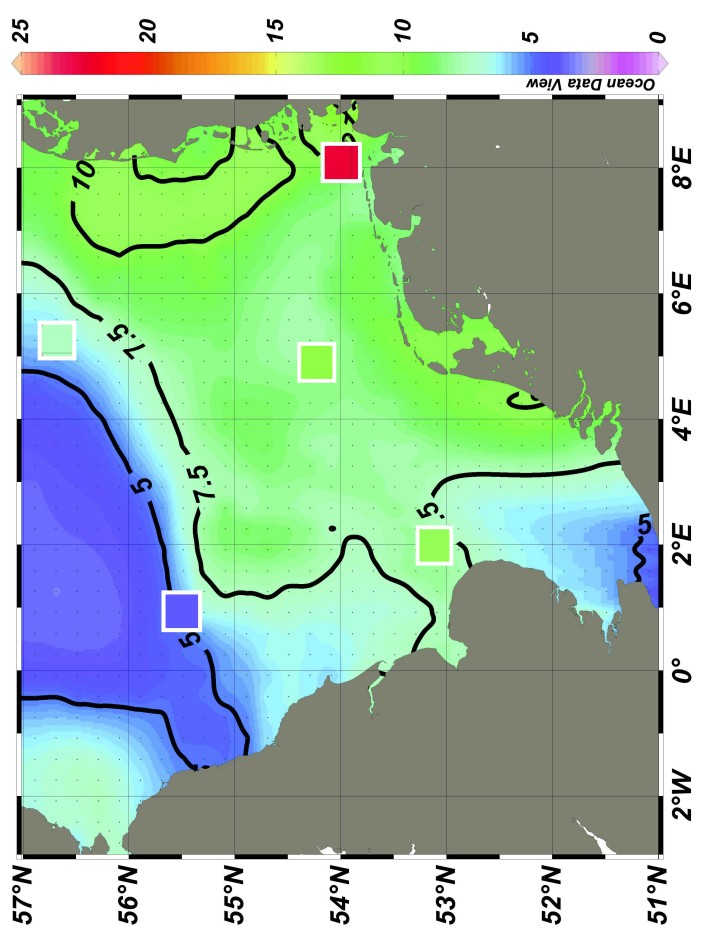



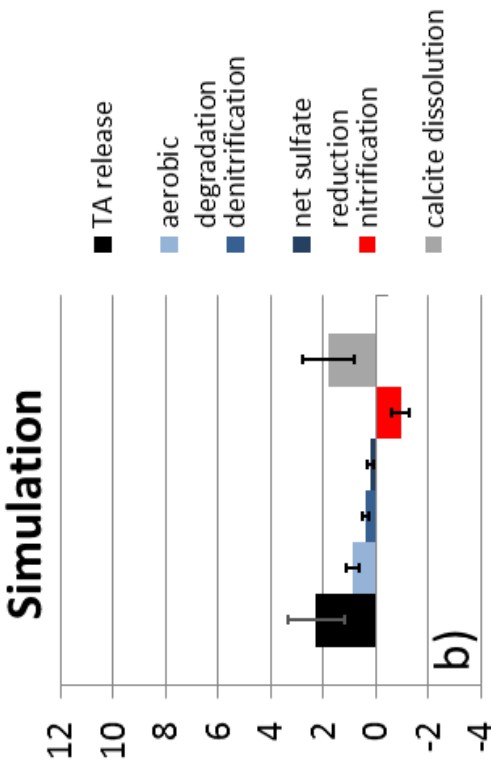

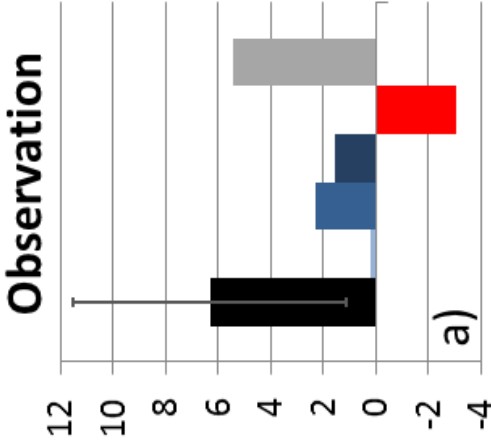

Fig. 4





Fig. 5





Fig. 6



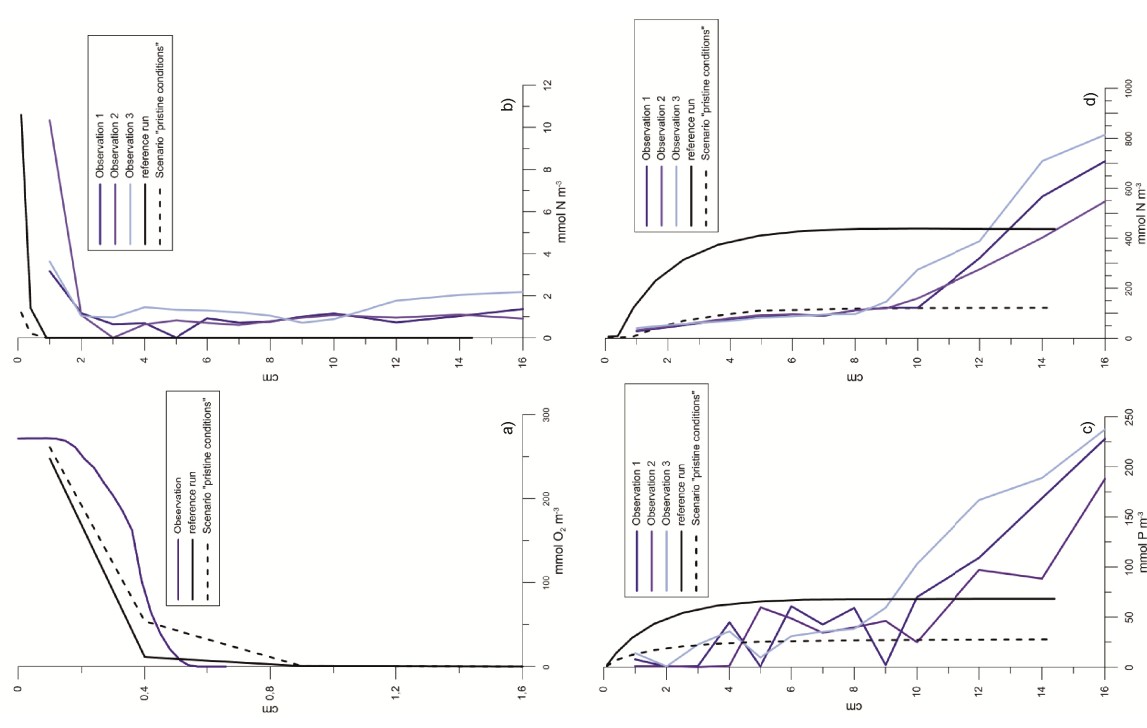

Fig. 7



Fig. 8





Fig. 9




Fig. 10

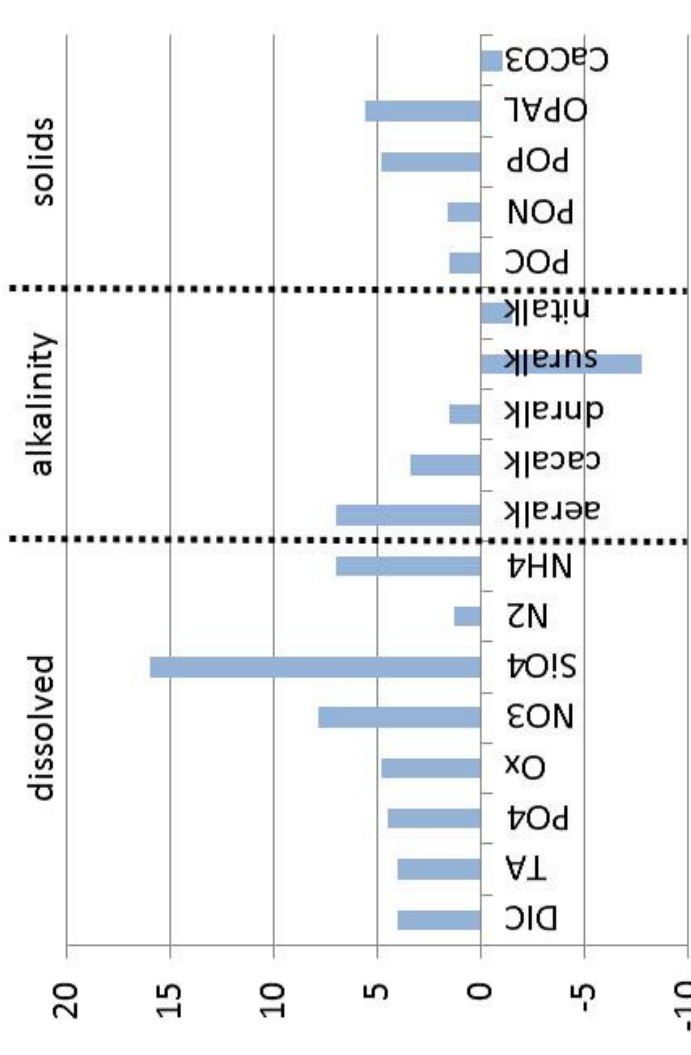



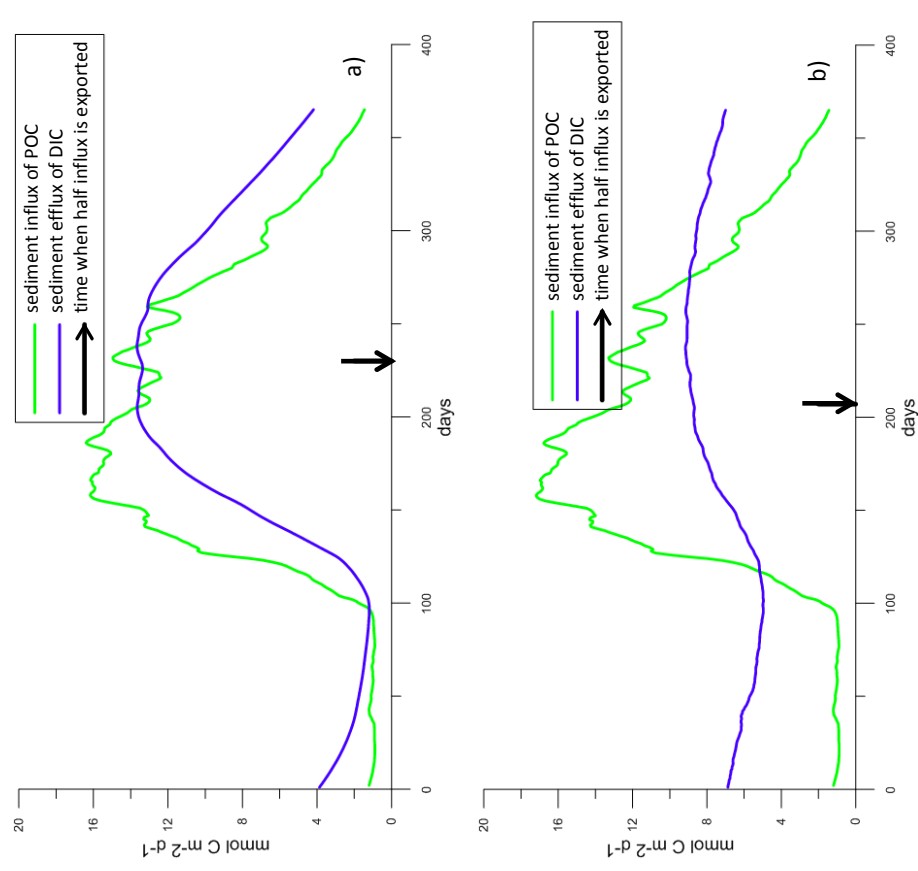

Fig. 11