# Peer review of "Interannual sedimentary effluxes of alkalinity in the southern North Sea: Model results compared with summer observations"

_Biogeosciences, 2018_

## Referee Comment (RC1) · Anonymous Referee #1 · 5 Mar 2018

GENERIC COMMENT the paper presents an interesting implementation of a vertically structured benthic model to estimate the alkalinity fluxes from the Southern North Sea sediments. This is a challenging topic that needs to be addressed and authors are commended for this. The methods are generally sound, with some clarification needed and some suggestions for improvements provided. The results are presented clearly, the discussion could benefit of some more in-depth analysis, particularly on the role of pelagic primary productivity and on the relevance of the alkalinity fluxes for the entire ecosystems.

SPECIFIC COMMENTS: section 2.2.5 and table 1: why I appreciate that turnover rate

for deep ocean would not be suitable for shelf seas, authors did not specify how they define the new values, if via calibration (against which observations?) or with literature (which references?).

Lines 144-146: authors assumed that in advective sediments the coefficient for diffusion is increased tenfold. They state that this has been determined by several sensitivity analysis, but they did not state what were the criteria of the sensitivity analysis (stability? calibration? Something else?) more details are needed

lines 175-178: authors claims that a reduction of 10% of riverine input of nitrogen corresponds to a "pristine" scenario without anthropogenic influence. Authors cite a paper to corroborate this assumption. However it seems to me that 10% is a bit of an underestimation for such and industrialised area.

Line 179-183: do the rates in the "plate run" scenario are comparable with those of table 1? I appreciate that the equation will be different and therefore the value of the parameter, but reporting these for a comparison would help in understanding how much of the difference is due to model structure and how much to simple parameter values

line 220 and following. I'm not sure that providing a point-to-point comparison on a single day is the more effective way to assess the model. Small shift in phenology (not rare in coupled biogeochemical models) could result in a significant error that could not be related to the benthic model rather to error in the physics or in the forcing. I would suggest to compare the observations with a longer temporal means (monthly?) and to discuss the uncertainty. Also, while visual comparison can be appealing, they are not much informative: I suggest to provide also measures of the actual fit. For example, in relative error term, I'm not sure figure 6 shows a much better fit.

Section 4.2: authors seem to suggest that the strong undersaturation of pCO2 in the German bight is driven by the strong alkalinity fluxes from benthos. I'm not entirely sure that the simple co-location of the two is enough to establish a causal link. For instance what's the role of pelagic primary production (PP)? A high PP could explain

both the strong undersaturation (DIC is fixed into plankton) and the alkalinity fluxes (due to strong POM settling and associated processes in the benthic environment). Have the author tested to turn off the benthic fluxes of TA and check the consequences in the delta pCO2 signal?

Section 4.3: authors said that it's astonishing that the model simulates higher benthic-pelagic fluxes under higher porosity, when the diffusion coefficient is lower. Do authors have any suggestion on what are the mechanisms driving this?

Section 4.4: this section is important to understand the need for detailed model. However authors simply state the difference between the two models, without trying to tease out the reason behind that, particularly in regard to the difference in the seasonal signal

technical comment: please translate "gedankenexperiment" in English

---

## Referee Comment (RC2) · Anonymous Referee #2 · 28 Mar 2018

Review of the paper "Interannual sedimentary effluxes of alkalinity in the southern North Sea: Model results compared with summer observations" by J. Pätsch et al., submitted for possible publication to Biogeosciences.

The paper by Pätsch and coworkers deals with an important topic, namely on how coastal systems respond to increased nutrients input. Hitherto the focus of this discussion has largely been on the interaction between primary production, aerobic respiration and oxygen conditions with respect to changes of the overall $CO_2$ pool. In recent times this focus has been broadened to further consider anaerobic metabolic pathways for the respiration of organic matter, which in turn affect both the overall $CO_2$

and and also alkalinity pools. The paper by Pätsch et al. analyses this issue for the shallow waters of the North Sea and provides compelling evidence for the dependency of metabolic alkalinity generation on nutrient runoff/input. Overall I think the paper is publishable, as it constitutes a major step forward in this field. I do have some remarks for consideration to improve the paper.

Specific remarks:

Figures: All figures should be designed that each panel does clearly reveal the property, its unit and magnitude, such the the reader can read the panel without referring back to the caption. In the present form, only Fig. 9 has been designed accordingly. The other figures are not usable without the caption, which makes it very difficult to follow the text, and might be in part even misleading, such as in Fig. 7, where the property (N) is given, but points to different species of the N cycle. I think this point is also important if any colleagues might use such a figure for a talk or teaching. The only way to prevent misunderstandings is to carry the information in each panel as well (even if redundant).

Page 2, line 4: loss of reduced sulfate products: this (mainly?) refers to burial of sulfides (FeS, FeS2). If so, please be specific.

Page 8, section 3.2, and related figures. While technically correct, it appears during the first and second read confusing to attribute aerobic degradation an alkalinity gain, and then separately name nitrification to a reduction of alkalinity. As this route is o pursued throughout the paper, please explicitly explain this here. Also a consideration might be to lump them together, if these processes always occur together. In other words, are there situations in this modelling study, where these two processes are not strictly coupled, which in turn would justify the need to treat them separately?

Page 9 lines 212-218. Calcium carbonate. To me the calcium carbonate discussion reads a bit like a closed loop argument, since apparently all hinges on the prescribed POC:PIC ratio. I have problems to see the need and justification for this discussion

as it appears to be arbitrarily(!) controlled by the choice to the POC:PIC constant. I'd recommended to tone down this discussion, and focus it on the point where it might be necessary, namely when attempting to explain the difference outcomes of observational (summertime) studies, and year-round modeling studies, the latter ones possibly closing the CaCO3 budget. Alternatively, an attempt could be made to adjust/establish that ratio to improve the pCO2 fields, as for example provided by Thomas et al. 2004. Another weakness of the fixed POC:PIC ratio is that CaCO3 production is reported to occur sporadically, and not necessarily in tandem with primary production, which in turn would diminish or vanish the assumed advantage over the observational records.

Page 11, lines 282-284: I think this statement could be strengthened by looking into coherence or lag-times between changes in NO3-runoff and extent of denitrification. The data are there, so an analysis in that direction should be easy to be carried out.

Page 11. lines 285-300.

I think this discussion needs to be rewritten to some degree, as I see major arguments missing, or not fully considered.

Why attempting to relate a 9-month accumulation to an in-situ observation of a seasonally varying property? While I can can see the reason for this, yet it has to be considered:

A: the residence time of the water at any given location. If the residence time is on average much shorter than the 9-month integration, the latter one does not make sense and should be shortened.

B: the entire concept only applies to vertically mixed regions. In stratified regions the zone of AT production is separate from the surface, thus if there is any impact it can be only visible, once, stratification breaks down in autumn (if at all). Also there are regions in the North Sea, which are permanently stratified. See for example Burt et al. 2014 (GBC) and observations of shortlived Ra isotopes in North Sea surface waters.

[Figure]

C: for the more Northern regions: what about transport times scales and amounts? Is any of that alkalinity produced in the southern surface sediments transported northward and might have an effect on the pCO2 there? I am not sure whether this plays a role, but it might be more likely to occur than the vertical impact mentioned under B. (See Burt et al., 2016 L&O)

---

## Author Comment (AC1) · 17 Apr 2018

Thank you for the constructive review of our manuscript. We answered your questions and followed your advices whenever possible. In the following lines we copied your text as bold and gave our remarks in detail.

**GENERIC COMMENT the paper presents an interesting implementation of a vertically structured benthic model to estimate the alkalinity fluxes from the Southern North Sea sediments. This is a challenging topic that needs to be addressed and authors are commended for this. The methods are generally sound, with some clarification needed and some suggestions for improvements provided. The results are presented clearly, the discussion could benefit of some more in-depth analysis, particularly on the role of pelagic primary productivity and on the relevance of the alkalinity fluxes for the entire ecosystems.**

Thank you for the hints regarding primary production and the role of alkalinity fluxes. During revising the text and answering your specific comments primary production was highlighted. One example is the positive feedback of pelagic production when enhanced nitrogen effluxes occur. In our model the direct effect of alkalinity effluxes on the whole ecosystem is restricted to the pelagic carbonate system and the air-sea flux of $CO_2$. The indirect effect of connected effluxes of nutrients and oxygen is discussed now in a more comprehensive manner.

**SPECIFIC COMMENTS: section 2.2.5 and table 1: why I appreciate that turnover rate for deep ocean would not be suitable for shelf seas, authors did not specify how they define the new values, if via calibration (against which observations?) or with literature (which references?).**

As for the North Sea the annual budget of carbon export into the sediment and the efflux of DIC is nearly in balance, the changes of the reaction rates aimed at this target. In a first step we replaced the spatial uniform porosity used in the open ocean sediment module (0.85 first layer) to observation-based values for the southern North Sea (between 0.3 – 0.51). Additionally, the constant diffusion rate was replaced by a porosity, temperature and substrate dependent formulation. These changes made it necessary to adjust the turnover rates. As shelf areas are hardly resolved in the coarse resolution of the global model seasonality of organic matter fluxes and DIC effluxes were no tuning criteria. We changed the text in section 2.3.5 accordingly.

**Lines 144-146: authors assumed that in advective sediments the coefficient for diffusion is increased tenfold. They state that this has been determined by several sensitivity analysis, but they did not state what were the criteria of the sensitivity analysis (stability? calibration? Something else?) more details are needed**

As said before the target was to achieve a more or less equal annual DIC efflux and POC influx. Furthermore, the seasonality of the DIC efflux should resemble the known temporal evolution. An upper constraint for the diffusion coefficients was the penetration depth of significant oxygen concentrations. Below 1 cm depth almost no oxygen should be detected. We changed the text accordingly.

**lines 175-178: authors claims that a reduction of 10% of riverine input of nitrogen corresponds to a "pristine" scenario without anthropogenic influence. Authors cite a paper to corroborate this**

**assumption. However it seems to me that 10% is a bit of an underestimation for such and industrialised area.**

This is a misunderstanding. We reduced the riverine input to 10 % of the anthropogenic value.

**Line 179-183: do the rates in the "plate run" scenario are comparable with those of table 1? I appreciate that the equation will be different and therefore the value of the parameter, but reporting these for a comparison would help in understanding how much of the difference is due to model structure and how much to simple parameter values**

The carbon degradation rate of the plate run is defined as a time constant (2.8E-2/d), whereas the aerobic rate of the reference run is an oxygen dependent rate (r1 =2 E-10 m3/(mmol [O2] s). For an off-shore station (54.4 °N, 7.4 °E) with a typical oxygen concentration of 100 mmol/m3 r1 results in a comparable rate ( 1.7 E-3 /d). In contrast to the plate run where nearly all POC is dissolved over one year POC concentrations in the upper most sediment level of about 0.35 E6 mmol C/m3 in winter and 0.48 E6 mmol C/m3 in summer are found in the reference run. This results in a winter flux of 1.2 mmol C/(d m2) and a summer flux of 1.6 mmol C/(d m2). The corresponding fluxes of the plate run were 0.36 mmol C/(d m2) in winter and 2.3 mmol C/(d m2) in summer. We added some text in section 4.4 accordingly.

**line 220 and following. I'm not sure that providing a point-to-point comparison on a single day is the more effective way to assess the model. Small shift in phenology (not rare in coupled biogeochemical models) could result in a significant error that could not be related to the benthic model rather to error in the physics or in the forcing. I would suggest to compare the observations with a longer temporal means (monthly?) and to discuss the uncertainty. Also, while visual comparison can be appealing, they are not much informative: I suggest to provide also measures of the actual fit. For example, in relative error term, I'm not sure figure 6 shows a much better fit.**

We switched from the analysis of 15 September to September means. But this did not change much because especially in September no big changes were expected (compare Fig 11b). We added an error analysis (definition in section 2.6).

**Section 4.2: authors seem to suggest that the strong undersaturation of pCO2 in the German bight is driven by the strong alkalinity fluxes from benthos. I'm not entirely sure that the simple co-location of the two is enough to establish a causal link. For instance what's the role of pelagic primary production (PP)? A high PP could explain both the strong undersaturation (DIC is fixed into plankton) and the alkalinity fluxes (due to strong POM settling and associated processes in the benthic environment). Have the author tested to turn off the benthic fluxes of TA and check the consequences in the delta pCO2 signal?**

This was a very helpful hint: in a further sensitivity run we switched off the benthic TA fluxes in the reference run. Consequently the pCO2 values increased. But the coastal low pCO2 did not fully vanish. We concluded that both the primary production and the benthic TA effluxes were responsible for the near-coast low pCO2 values. We added some text accordingly.

**Section 4.3: authors said that it's astonishing that the model simulates higher benthic pelagic fluxes under higher porosity, when the diffusion coefficient is lower. Do authors have any suggestion on what are the mechanisms driving this?**

To understand this contra-intuitive dynamics we compared the model results of the high porosity run (HP with 0.51) with the low porosity run (LP with 0.3) in the first and second spinup year. At the beginning of the first spinup year all conditions are the same. Until spring the flux of oxygen into the sediment was lower in HP because there the effective diffusivity was lower than in LP. The lower oxygen content in HP stimulated the benthic anaerobic processes. At the end of the first year this resulted into a higher efflux of NH4 from the sediment in the HP scenario. The higher NH4 efflux of the HP scenario was not compensated by the higher NO3 flux into the sediment. At the end of the year more DIN was in the pelagic water column in the HP scenario than in the LP scenario. In the second year this surplus of DIN stimulated higher primary production for the HP scenario. The corresponding enhanced particle export additionally increased the benthic-pelagic fluxes. The loss of N2 due to enhanced denitrification was compensated by the larger NH4 efflux. These deviating dynamics are even stronger at stations with lower pelagic DIN concentrations. We added some text accordingly.

**Section 4.4: this section is important to understand the need for detailed model. However authors simply state the difference between the two models, without trying to tease out the reason behind that, particularly in regard to the difference in the seasonal signal**

There are several reasons for the deviating seasonal cycle of DIC efflux. In general the less pronounced cycle comes about
- the structure of the 3d-sediment model which leads to a combination of multiple time scales acting on the reaction rates due to diffusive processes between the layers,
- the fact that the remineralization fluxes do not produce immediately effluxes. In case of the 3d-sediment model the dissolved compounds have to be transferred via diffusion into the pelagic system,
- the reservoir effect in the 3d-sediment model: Whereas the 2d-plate model more or less all POC is degraded after winter, in the 3d-sediment model a relative high POC concentration remains.

We added text accordingly.

**technical comment: please translate "gedankenexperiment" in English.**

done

---

## Author Comment (AC2) · 17 Apr 2018

We would like to thank for the constructive review of our manuscript. We answered your questions and followed your advices whenever possible. In the following lines we copied your text as bold and gave our remarks in detail.

**Review of the paper "Interannual sedimentary effluxes of alkalinity in the southern North Sea: Model results compared with summer observations" by J. Pätsch et al., submitted for possible publication to Biogeosciences.**

**The paper by Pätsch and coworkers deals with an important topic, namely on how coastal systems respond to increased nutrients input. Hitherto the focus of this discussion has largely been on the interaction between primary production, aerobic respiration and oxygen conditions with respect to changes of the overall CO2 pool. In recent times this focus has been broadened to further consider anaerobic metabolic pathways for the respiration of organic matter, which in turn affect both the overall CO2 and and also alkalinity pools. The paper by Pätsch et al. analyses this issue for the shallow waters of the North Sea and provides compelling evidence for the dependency of metabolic alkalinity generation on nutrient runoff/input. Overall I think the paper is publishable, as it constitutes a major step forward in this field. I do have some remarks for consideration to improve the paper.**

**Specific remarks:**
**Figures: All figures should be designed that each panel does clearly reveal the property, its unit and magnitude, such the the reader can read the panel without referring back to the caption. In the present form, only Fig. 9 has been designed accordingly. The other figures are not usable without the caption, which makes it very difficult to follow the text, and might be in part even misleading, such as in Fig. 7, where the property (N) is given, but points to different species of the N cycle. I think this point is also important if any colleagues might use such a figure for a talk or teaching. The only way to prevent misunderstandings is to carry the information in each panel as well (even if redundant).**

done

**Page 2, line 4: loss of reduced sulfate products: this (mainly?) refers to burial of sulfides (FeS, FeS2). If so, please be specific.**

We specified the products.

**Page 8, section 3.2, and related figures. While technically correct, it appears during the first and second read confusing to attribute aerobic degradation an alkalinity gain, and then separately name nitrification to a reduction of alkalinity. As this route is o pursued throughout the paper, please explicitly explain this here. Also a consideration might be to lump them together, if these processes always occur together. In other words, are there situations in this modelling study, where these two processes are not strictly coupled, which in turn would justify the need to treat them separately?**

We introduced a sentence in section 2.3.3. By definition x mol TA is produced by incomplete aerobic degradation to ammonium and 2x mol TA are consumed by nitrification. Fig. 4b shows that TA

generation and consumption by these two processes is about equal in the sediment. This clearly shows that the two processes are not strictly coupled.

**Page 9 lines 212-218. Calcium carbonate. To me the calcium carbonate discussion reads a bit like a closed loop argument, since apparently all hinges on the prescribed POC:PIC ratio. I have problems to see the need and justification for this discussion as it appears to be arbitrarily(!) controlled by the choice to the POC:PIC constant. I' d recommended to tone down this discussion, and focus it on the point where it might be necessary, namely when attempting to explain the difference outcomes of observational (summertime) studies, and year-round modeling studies, the latter ones possibly closing the CaCO3 budget. Alternatively, an attempt could be made to adjust/establish that ratio to improve the pCO2 fields, as for example provided by Thomas et al. 2004. Another weakness of the fixed POC:PIC ratio is that CaCO3 production is reported to occur sporadically, and not necessarily in tandem with primary production, which in turn would diminish or vanish the assumed advantage over the observational records.**

We toned done the discussion and gave the hint to the possibility of sporadically occurring calcite production.

**Page 11, lines 282-284: I think this statement could be strengthened by looking into coherence or lag-times between changes in NO3-runoff and extent of denitrification. The data are there, so an analysis in that direction should be easy to be carried out.**

Thank you for this advice. We calculated the correlation coefficient for different monthly time shifts. And indeed a shift of 2-3 months gave the best (and significant) correlation. We added corresponding text in section 4.1.

**Page 11. lines 285-300.**
**I think this discussion needs to be rewritten to some degree, as I see major arguments missing, or not fully considered.**

**Why attempting to relate a 9-month accumulation to an in-situ observation of a seasonally varying property? While I can can see the reason for this, yet it has to be considered:**

**A: the residence time of the water at any given location. If the residence time is on average much shorter than the 9-month integration, the latter one does not make sense and should be shortened.**

**B: the entire concept only applies to vertically mixed regions. In stratified regions the zone of AT production is separate from the surface, thus if there is any impact it can be only visible, once, stratification breaks down in autumn (if at all). Also there are regions in the North Sea, which are permanently stratified. See for example Burt et al. 2014 (GBC) and observations of shortlived Ra isotopes in North Sea surface waters.**

**C: for the more Northern regions: what about transport times scales and amounts? Is any of that alkalinity produced in the southern surface sediments transported northward and might have an**

**effect on the pCO2 there? I am not sure whether this plays a role, but it might be more likely to occur than the vertical impact mentioned under B. (See Burt et al., 2016 L&O)**

We understand that our approach has several shortcomings. One of them is the arbitrariness of choosing the 15$^{th}$ of September.  Therefore we recalculated the pCO2 as September mean. Another shortcoming is the fact that for some places the alkalinity effluxes have no impact on local pCO2, neither for September nor for any other time. To overcome this we analyzed the temporal cumulated air-sea flux of CO2 from the beginning of the year until mid of September which at least weakens the argument related to flushing times. The attached figure shows horizontal distributions of the cumulated fluxes for both model runs. The patter resembles the ΔpCO2 distributions in September (Fig. 9). Northern areas with greater water depths show hardly any change. Small differences can be seen in the southern open sea areas and high differences occur near the continental coast where also the differences due to altered primary production were found. We added corresponding text in section 4.2.

Reference run
Cumulative ASF until
mid of September
(mmol m$^{-2}$)

[Figure]

Pristine condition run
Cumulative ASF until
mid of September
(mmol m$^{-2}$)